# Towards Generalization and Simplicity in Continuous Control

**Aravind Rajeswaran**[*]    **Kendall Lowrey**[*]    **Emanuel Todorov**    **Sham Kakade**

University of Washington Seattle

{ aravraj, klowrey, todorov, sham } @ cs.washington.edu

## Abstract

This work shows that policies with simple linear and RBF parameterizations can be trained to solve a variety of widely studied continuous control tasks, including the gym-v1 benchmarks. The performance of these trained policies are competitive with state of the art results, obtained with more elaborate parameterizations such as fully connected neural networks. Furthermore, the standard training and testing scenarios for these tasks are shown to be very limited and prone to over-fitting, thus giving rise to only trajectory-centric policies. Training with a diverse initial state distribution induces more global policies with better generalization. This allows for interactive control scenarios where the system recovers from large on-line perturbations; as shown in the supplementary video.

## 1 Introduction

Deep reinforcement learning (deepRL) has recently achieved impressive results on a number of hard problems, including sequential decision making in game domains [1, 2]. This success has motivated efforts to adapt deepRL methods for control of physical systems, and has resulted in rich motor behaviors [3, 4]. The complexity of systems solvable with deepRL methods is not yet at the level of what can be achieved with trajectory optimization (planning) in simulation [5, 6, 7], or with hand-crafted controllers on physical robots (e.g. Boston Dynamics). However, RL approaches are exciting because they are generic, model-free, and highly automated.

Recent success of RL [2, 8, 9, 10, 11] has been enabled largely due to engineering efforts such as large scale data collection [1, 2, 11] or careful systems design [8, 9] with well behaved robots. When advances in a field are largely empirical in nature, it is important to understand the relative contributions of representations, optimization methods, and task design or modeling: both as a sanity check and to scale up to harder tasks. Furthermore, in line with Occam's razor, the simplest reasonable approaches should be tried and understood first. A thorough understanding of these factors is unfortunately lacking in the community.

In this backdrop, we ask the pertinent question: "What are the simplest set of ingredients needed to succeed in some of the popular benchmarks?" To attempt this question, we use the Gym-v1 [12] continuous control benchmarks, which have accelerated research and enabled objective comparisons. Since the tasks involve under-actuation, contact dynamics, and are high dimensional (continuous space), they have been accepted as benchmarks in the deepRL community. Recent works test their algorithms either exclusively or primarily on these tasks [13, 4, 14], and success on these tasks have been regarded as demonstrating a "proof of concept".

**Our contributions:**    Our results and their implications are highlighted below with more elaborate discussions in Section 5:

---

[*] Equal contributions. Project page: https://sites.google.com/view/simple-pol

1. The success of recent RL efforts to produce rich motor behaviors have largely been attributed to the use of multi-layer neural network architectures. This work is among the first to carefully analyze the role of representation, and our results indicate that very simple policies including linear and RBF parameterizations are able to achieve state of the art results on widely studied tasks. Furthermore, such policies, particularly the linear ones, can be trained significantly faster due to orders of magnitude fewer parameters. This indicates that even for tasks with complex dynamics, there could exist relatively simple policies. This opens the door for studying a wide range of representations in addition to deep neural networks, and understand trade-offs including computational time, theoretical justification, robustness, sample complexity etc.

2. We study these issues not only with regards to the performance metric at hand but we also take the further step in examining them in the context of robustness. Our results indicate that, with conventional training methods, the agent is able to successfully learn a limit cycle for walking, but cannot recover from any perturbations that are delivered to it. For transferring the success of RL to robotics, such brittleness is highly undesirable.

3. Finally, we directly attempt to learn more robust policies through using more diverse training conditions, which favor such policies. This is similar in spirit to the model ensemble approaches [15, 16] and domain randomization approaches [17, 18], which have successfully demonstrated improved robustness and simulation to real world transfer. Under these new and more diverse training scenarios, we again find that there is no compelling evidence to favor the use of multi-layer architectures, at least for the benchmark tasks. On a side note, we also provide interactive testing of learned policies, which we believe is both novel and which sheds light on the robustness of trained policies.

## 2  Problem Formulation and Methods

We consider Markov Decision Processes (MDPs) in the average reward setting, which is defined using the tuple: $\mathcal{M} = \{\mathcal{S}, \mathcal{A}, \mathcal{R}, \mathcal{T}, \rho_0\}$. $\mathcal{S} \subseteq \mathbb{R}^n$, $\mathcal{A} \subseteq \mathbb{R}^m$, and $\mathcal{R} : \mathcal{S} \times \mathcal{A} \to \mathbb{R}$ are a (continuous) set of states, set of actions, and reward function respectively, and have the usual meaning. $\mathcal{T} : \mathcal{S} \times \mathcal{A} \to \mathcal{S}$ is the stochastic transition function and $\rho_0$ is the probability distribution over initial states. We wish to solve for a stochastic policy of the form $\pi : \mathcal{S} \times \mathcal{A} \to \mathbb{R}_+$, which optimizes the objective function:

$$\eta(\pi) = \lim_{T \to \infty} \frac{1}{T} \, \mathbb{E}_{\pi, \mathcal{M}} \left[ \sum_{t=1}^{T} r_t \right]. \tag{1}$$

Since we use simulations with finite length rollouts to estimate the objective and gradient, we approximate $\eta(\pi)$ using a finite $T$. In this finite horizon rollout setting, we define the value, $Q$, and advantage functions as follows:

$$V^\pi(s, t) = \mathbb{E}_{\pi, \mathcal{M}} \left[ \sum_{t'=t}^{T} r_{t'} \right] \qquad Q^\pi(s, a, t) = \mathbb{E}_{\mathcal{M}} \left[ \mathcal{R}(s, a) \right] + \mathbb{E}_{s' \sim \mathcal{T}(s, a)} \left[ V^\pi(s', t+1) \right]$$

$$A^\pi(s, a, t) = Q^\pi(s, a, t) - V^\pi(s, t)$$

Note that even though the value functions are time-varying, we still optimize for a stationary policy. We consider parametrized policies $\pi_\theta$, and hence wish to optimize for the parameters $(\theta)$. Thus, we overload notation and use $\eta(\pi)$ and $\eta(\theta)$ interchangeably.

### 2.1  Algorithm

Ideally, a controlled scientific study would seek to isolate the challenges related to architecture, task design, and training methods for separate study. In practice, this is not entirely feasible as the results are partly coupled with the training methods. Here, we utilize a straightforward natural policy gradient method for training. The work in [19] suggests that this method is competitive with most state of the art methods. We now discuss the training procedure.

Using the likelihood ratio approach and Markov property of the problem, the sample based estimate of the policy gradient is derived to be [20]:

$$\nabla_\theta \hat{\eta}(\theta) = g = \frac{1}{T} \sum_{t=0}^{T} \nabla_\theta \log \pi_\theta(a_t | s_t) \hat{A}^\pi(s_t, a_t, t) \tag{2}$$

**Algorithm 1** Policy Search with Natural Gradient

1: Initialize policy parameters to $\theta_0$
2: **for** $k = 1$ **to** $K$ **do**
3:     Collect trajectories $\{\tau^{(1)}, \ldots \tau^{(N)}\}$ by rolling out the stochastic policy $\pi(\cdot; \theta_k)$.
4:     Compute $\nabla_\theta \log \pi(a_t|s_t; \theta_k)$ for each $(s, a)$ pair along trajectories sampled in iteration $k$.
5:     Compute advantages $A_k^\pi$ based on trajectories in iteration $k$ and approximate value function $V_{k-1}^\pi$.
6:     Compute policy gradient according to (2).
7:     Compute the Fisher matrix (4) and perform gradient ascent (5).
8:     Update parameters of value function in order to approximate $V_k^\pi(s_t^{(n)}) \approx R(s_t^{(n)})$, where $R(s_t^{(n)})$ is the empirical return computed as $R(s_t^{(n)}) = \sum_{t'=t}^T \gamma^{(t'-t)} r_t^{(n)}$. Here $n$ indexes over the trajectories.
9: **end for**

Gradient ascent using this "vanilla" gradient is sub-optimal since it is not the steepest ascent direction in the metric of the parameter space [21, 22]. The steepest ascent direction is obtained by solving the following local optimization problem around iterate $\theta_k$:

$$\underset{\theta}{\text{maximize}} \quad g^T(\theta - \theta_k) \qquad \text{subject to} \quad (\theta - \theta_k)^T F_{\theta_k}(\theta - \theta_k) \leq \delta, \tag{3}$$

where $F_{\theta_k}$ is the Fisher Information Metric at the current iterate $\theta_k$. We estimate $F_{\theta_k}$ as

$$\hat{F}_{\theta_k} = \frac{1}{T} \sum_{t=0}^T \nabla_\theta \log \pi_\theta(a_t|s_t) \nabla_\theta \log \pi_\theta(a_t|s_t)^T, \tag{4}$$

as originally suggested by Kakade [22]. This yields the steepest ascent direction to be $\hat{F}_{\theta_k}^{-1} g$ and corresponding update rule: $\theta_{k+1} = \theta_k + \alpha \hat{F}_{\theta_k}^{-1} g$. Here $\alpha$ is the step-size or learning rate parameter. Empirically, we observed that choosing a fixed value for $\alpha$ or an appropriate schedule is difficult [23]. Thus, we use the normalized gradient ascent procedure, where the normalization is under the Fisher metric. This procedure can be viewed as picking a normalized step size $\delta$ as opposed to $\alpha$, and solving the optimization problem in (3). This results in the following update rule:

$$\theta_{k+1} = \theta_k + \sqrt{\frac{\delta}{g^T \hat{F}_{\theta_k}^{-1} g}} \, \hat{F}_{\theta_k}^{-1} g. \tag{5}$$

A dimensional analysis of these quantities reveal that $\alpha$ has the unit of return$^{-1}$ whereas $\delta$ is dimensionless. Though units of $\alpha$ are consistent with a general optimization setting where step-size has units of objective$^{-1}$, in these problems, picking a good $\alpha$ that is consistent with the scales of the reward was difficult. On the other hand, a constant normalized step size was numerically more stable and easier to tune: for all the results reported in this paper, the same $\delta = 0.05$ was used. When more than one trajectory rollout is used per update, the above estimators can be used with an additional averaging over the trajectories.

For estimating the advantage function, we use the GAE procedure [13]. This requires learning a function that approximates $V_k^\pi$, which is used to compute $A_k^\pi$ along trajectories for the update in (5). GAE helps with variance reduction at the cost of introducing bias, and requires tuning hyperparameters like a discount factor and an exponential averaging term. Good heuristics for these parameters have been suggested in prior work. The same batch of trajectories cannot be used for both fitting the value function baseline, and also to estimate $g$ using (2), since it will lead to overfitting and a biased estimate. Thus, we use the trajectories from iteration $k-1$ to fit the value function, essentially approximating $V_{k-1}^\pi$, and use trajectories from iteration $k$ for computing $A_k^\pi$ and $g$. Similar procedures have been adopted in prior work [19].

## 2.2 Policy Architecture

**Linear policy:** We first consider a linear policy that directly maps from the observations to the motor torques. We use the same observations as used in prior work which includes joint positions,

joint velocities, and for some tasks, information related to contacts. Thus, the policy mapping is $a_t \sim \mathcal{N}(Ws_t + b, \sigma)$, and the goal is to learn $W$, $b$, and $\sigma$. For most of these tasks, the observations correspond to the state of the problem (in relative coordinates). Thus, we use the term states and observations interchangeably. In general, the policy is defined with observations as the input, and hence is trying to solve a POMDP.

**RBF policy:** Secondly, we consider a parameterization that enriches the representational capacity using random Fourier features of the observations. Since these features approximate the RKHS features under an RBF Kernel [24], we call this policy parametrization the RBF policy. The features are constructed as:

$$y_t^{(i)} = \sin\left(\frac{\sum_j P_{ij}s_t^{(j)}}{\nu} + \phi^{(i)}\right), \tag{6}$$

where each element $P_{ij}$ is drawn from $\mathcal{N}(0, 1)$, $\nu$ is a bandwidth parameter chosen approximately as the average pairwise distances between different observation vectors, and $\phi$ is a random phase shift drawn from $U[-\pi, \pi]$. Thus the policy is $a_t \sim \mathcal{N}(Wy_t + b, \sigma)$, where $W$, $b$, and $\sigma$ are trainable parameters. This architecture can also be interpreted as a two layer neural network: the bottom layer is clamped with random weights, a sinusoidal activation function is used, and the top layer is finetuned. The principal purpose for this representation is to slightly enhance the capacity of a linear policy, and the choice of activation function is not very significant.

## 3 Results on OpenAI gym-v1 benchmarks

As indicated before, we train linear and RBF policies with the natural policy gradient on the popular OpenAI gym-v1 benchmark tasks simulated in MuJoCo [25]. The tasks primarily consist of learning locomotion gaits for simulated robots ranging from a swimmer to a 3D humanoid (23 dof).

Figure 1 presents the learning curves along with the performance levels reported in prior work using TRPO and fully connected neural network policies. Table 1 also summarizes the final scores, where "stoc" refers to the stochastic policy with actions sampled as $a_t \sim \pi_\theta(s_t)$, while "mean" refers to using mean of the Gaussian policy, with actions computed as $a_t = \mathbb{E}[\pi_\theta(s_t)]$. We see that the linear policy is competitive on most tasks, while the RBF policy can outperform previous results on five of the six considered tasks. Though we were able to train neural network policies that match the results reported in literature, we have used publicly available prior results for an objective comparison. Visualizations of the trained linear and RBF policies are presented in the supplementary video. Given the simplicity of these policies, it is surprising that they can produce such elaborate behaviors.

Table 2 presents the number of samples needed for the policy performance to reach a threshold value for reward. The threshold value is computed as $90\%$ of the final score achieved by the stochastic linear policy. We visually verified that policies with these scores are proficient at the task, and hence the chosen values correspond to meaningful performance thresholds. We see that linear and RBF policies are able to learn faster on four of the six tasks.

All the simulated robots we considered are under-actuated, have contact discontinuities, and continuous action spaces making them challenging benchmarks. When adapted from model-based control [26, 5, 27] to RL, however, the notion of "success" established was not appropriate. To shape the behavior, a very narrow initial state distribution and termination conditions are used in the benchmarks. As a consequence, the learned policies become highly trajectory centric – i.e. they are good only where they tend to visit during training, which is a very narrow region. For example, the walker can walk very well when initialized upright and close to the walking limit cycle. Even small perturbations, as shown in the supplementary video, alters the visitation distribution and dramatically degrades the policy performance. This makes the agent fall down at which point it is unable to get up. Similarly, the swimmer is unable to turn when its heading direction is altered. For control applications, this is undesirable. In the real world, there will always be perturbations – stochasticity in the environment, modeling errors, or wear and tear. Thus, the specific task design and notion of success used for the simulated characters are not adequate. However, the simulated robots themselves are rather complex and harder tasks could be designed with them, as partly illustrated in Section 4.

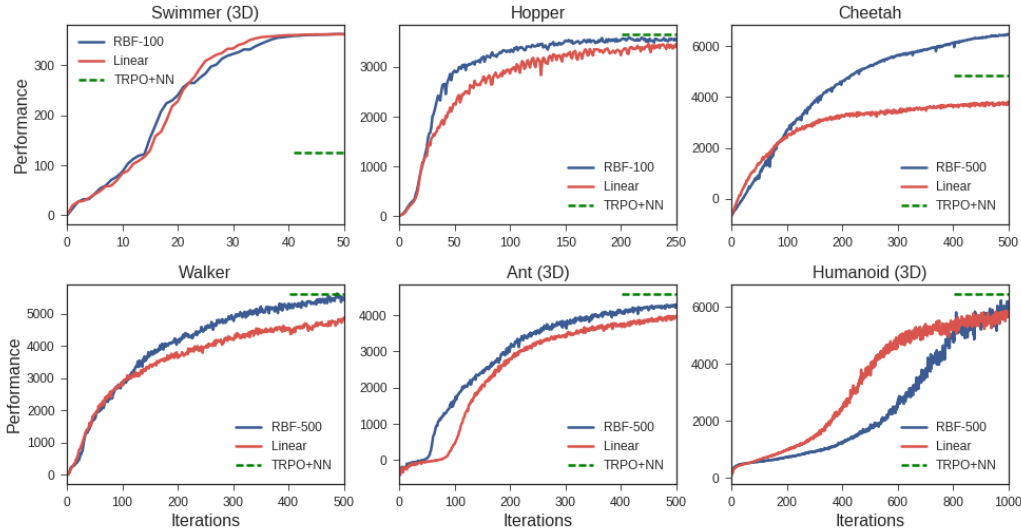

Figure 1: Learning curves for the Linear and RBF policy architectures. The green line corresponding to the reward achieved by neural network policies on the OpenAI Gym website, as of 02/24/2017 (trained with TRPO). It is observed that for all the tasks, linear and RBF parameterizations are competitive with state of the art results. The learning curves depicted are for the stochastic policies, where the actions are sampled as $a_t \sim \pi_\theta(s_t)$. The learning curves have been averaged across three runs with different random seeds.

Table 1: Final performances of the policies

| Task | Linear | | RBF | | NN |
|------|--------|--------|--------|--------|--------|
| | stoc | mean | stoc | mean | TRPO |
| Swimmer | 362 | **366** | 361 | 365 | 131 |
| Hopper | 3466 | 3651 | 3590 | **3810** | 3668 |
| Cheetah | 3810 | 4149 | 6477 | **6620** | 4800 |
| Walker | 4881 | 5234 | 5631 | **5867** | 5594 |
| Ant | 3980 | 4607 | 4297 | 4816 | **5007** |
| Humanoid | 5873 | 6440 | 6237 | **6849** | 6482 |

Table 2: Number of episodes to achieve threshold

| Task | Th. | Linear | RBF | TRPO+NN |
|------|------|--------|--------|---------|
| Swimmer | 325 | **1450** | 1550 | N-A |
| Hopper | 3120 | 13920 | **8640** | 10000 |
| Cheetah | 3430 | 11250 | 6000 | **4250** |
| Walker | 4390 | 36840 | 25680 | **14250** |
| Ant | 3580 | 39240 | **30000** | 73500 |
| Humanoid | 5280 | **79800** | 96720 | 87000 |

# 4 Modified Tasks and Results

Using the same set of simulated robot characters outlined in Section 3, we designed new tasks with two goals in mind: (a) to push the representational capabilities and test the limits of simple policies; (b) to enable training of "global" policies that are robust to perturbations and work from a diverse set of states. To this end, we make the following broad changes, also summarized in Table 3:

1. Wider initial state distribution to force generalization. For example, in the walker task, some fraction of trajectories have the walker initialized prone on the ground. This forces the agent to simultaneously learn a get-up skill and a walk skill, and not forget them as the learning progresses. Similarly, the heading angle for the swimmer and ant are randomized, which encourages learning of a turn skill.

2. Reward shaping appropriate with the above changes to the initial state distribution. For example, when the modified swimmer starts with a randomized heading angle, we include a small reward for adjusting its heading towards the correct direction. In conjunction, we also remove all termination conditions used in the Gym-v1 benchmarks.

3. Changes to environment's physics parameters, such as mass and joint torque. If the agent has sufficient power, most tasks are easily solved. By reducing an agent's action ability and/or increasing its mass, the agent is more under-actuated. These changes also produce more realistic looking motion.

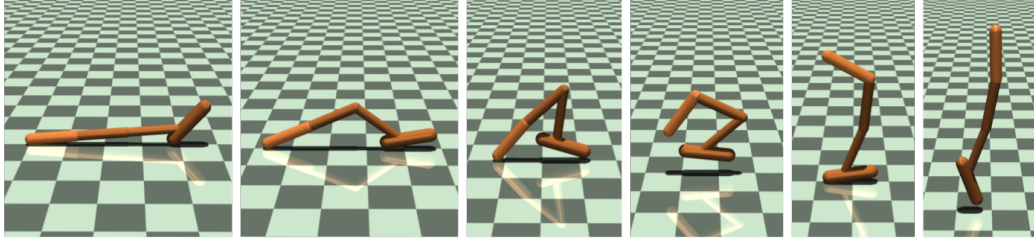

Figure 2: Hopper completes a get-up sequence before moving to its normal forward walking behavior. The getup sequence is learned along side the forward hopping in the modified task setting.

Table 3: Modified Task Description

$v_x$ is forward velocity; $\theta$ is the heading angle; $p_z$ is the height of torso; and $a$ is the action.

| Task | Description | Reward (des = desired value) |
|------|-------------|------------------------------|
| Swimmer (3D) | Agent swims in the desired direction. Should recover (turn) if rotated around. | $v_x - 0.1|\theta - \theta^{des}| - 0.0001||a||^2$ |
| Hopper (2D) | Agent hops forward as fast as possible. Should recover (get up) if pushed down. | $v_x - 3||p_z - p_z^{des}||^2 - 0.1||a||^2$ |
| Walker (2D) | Agent walks forward as fast as possible. Should recover (get up) if pushed down. | $v_x - 3||p_z - p_z^{des}||^2 - 0.1||a||^2$ |
| Ant (3D) | Agent moves in the desired direction. Should recover (turn) if rotated around. | $v_x - 3||p_z - p_z^{des}||^2 - 0.01||a||^2$ |

Combined, these modifications require that the learned policies not only make progress towards maximizing the reward, but also recover from adverse conditions and resist perturbations. An example of this is illustrated in Figure 4, where the hopper executes a get-up sequence before hopping to make forward progress. Furthermore, at test time, a user can interactively apply pushing and rotating perturbations to better understand the failure modes. We note that these interactive perturbations may not be the ultimate test for robustness, but a step towards this direction.

**Representational capacity** The supplementary video demonstrates the trained policies. We concentrate on the results of the walker task in the main paper. Figure 3 studies the performance as we vary the representational capacity. Increasing the Fourier features allows for more expressive policies and consequently allow for achieving a higher score. The policy with 500 Fourier features performs the best, followed by the fully connected neural network. The linear policy also makes forward progress and can get up from the ground, but is unable to learn as efficient a walking gait.

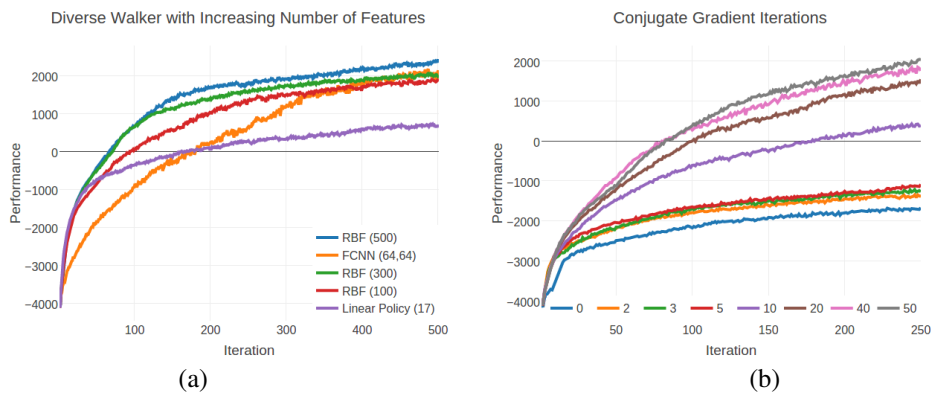

Figure 3: (a) Learning curve on modified walker (diverse initialization) for different policy architectures. The curves are averaged over three runs with different random seeds. (b) Learning curves when using different number of conjugate gradient iterations to compute $\hat{F}_{\theta_k}^{-1} g$ in (5). A policy with 300 Fourier features has been used to generate these results.

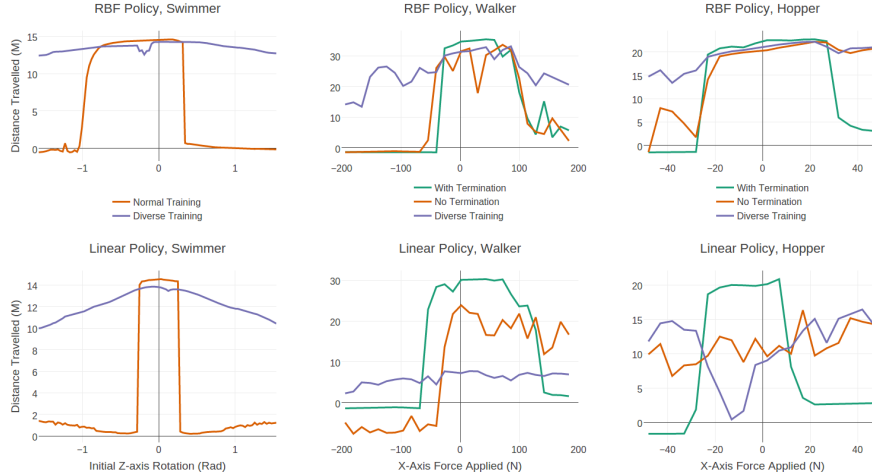

Figure 4: We test policy robustness by measuring distanced traveled in the swimmer, walker, and hopper tasks for three training configurations: (a) with termination conditions; (b) no termination, and peaked initial state distribution; and (c) with diverse initialization. Swimmer does not have a termination option, so we consider only two configurations. For the case of swimmer, the perturbation is changing the heading angle between $-\pi/2.0$ and $\pi/2.0$, and in the case of walker and hopper, an external force for 0.5 seconds along its axis of movement. All agents are initialized with the same positions and velocities.

**Perturbation resistance**   Next, we test the robustness of our policies by perturbing the system with an external force. This external force represents an unforeseen change which the agent has to resist or overcome, thus enabling us to understand push and fall recoveries. Fall recoveries of the trained policies are demonstrated in the supplementary video. In these tasks, perturbations are not applied to the system during the training phase. Thus, the ability to generalize and resist perturbations come entirely out of the states visited by the agent during training. Figure 4 indicates that the RBF policy is more robust, and also that diverse initializations are important to obtain the best results. This indicates that careful design of initial state distributions are crucial for generalization, and to enable the agent to learn a wide range of skills.

# 5   Summary and Discussion

The experiments in this paper were aimed at trying to understand the effects of (a) representation; (b) task modeling; and (c) optimization. We summarize the results with regard to each aforementioned factor and discuss their implications.

**Representation**   The finding that linear and RBF policies can be trained to solve a variety of continuous control tasks is very surprising. Recently, a number of algorithms have been shown to successfully solve these tasks [3, 28, 4, 14], but all of these works use multi-layer neural networks. This suggests a widespread belief that expressive function approximators are needed to capture intricate details necessary for movements like running. The results in this work conclusively demonstrates that this is not the case, at least for the limited set of popular testbeds. This raises an interesting question: what are the capability limits of shallow policy architectures? The linear policies were not exemplary in the "global" versions of the tasks, but it must be noted that they were not terrible either. The RBF policy using random Fourier features was able to successfully solve the modified tasks producing global policies, suggesting that we do not yet have a sense of its limits.

**Modeling**   When using RL methods to solve practical problems, the world provides us with neither the initial state distribution nor the reward. Both of these must be designed by the researcher and must be treated as assumptions about the world or prescriptions about the required behavior. The quality of assumptions will invariably affect the quality of solutions, and thus care must be taken in this process. Here, we show that starting the system from a narrow initial state distribution produces

elaborate behaviors, but the trained policies are very brittle to perturbations. Using a more diverse state distribution, in these cases, is sufficient to train robust policies.

**Optimization** In line with the theme of simplicity, we first tried to use REINFORCE [20], which we found to be very sensitive to hyperparameter choices, especially step-size. There are a class of policy gradient methods which use pre-conditioning to help navigate the warped parameter space of probability distributions and for step size selection. Most variants of pre-conditioned policy gradient methods have been reported to achieve state of the art performance, all performing about the same [19]. We feel that the used natural policy gradient method is the most straightforward pre-conditioned method. To demonstrate that the pre-conditioning helps, Figure 3 depicts the learning curve for different number of CG iterations used to compute the update in (5). The curve corresponding to $CG = 0$ is the REINFORCE method. As can be seen, pre-conditioning helps with the learning process. However, there is a trade-off with computation, and hence using an intermediate number of CG steps like 20 could lead to best results in wall-clock sense for large scale problems.

We chose to compare with neural network policies trained with TRPO, since it has demonstrated impressive results and is closest to the algorithm used in this work. Are function approximators linear with respect to free parameters sufficient for other methods is an interesting open question (in this sense, RBFs are linear but NNs are not). For a large class of methods based on dynamic programming (including Q-learning, SARSA, approximate policy and value iteration), linear function approximation has guaranteed convergence and error bounds, while non-linear function approximation is known to diverge in many cases [29, 30, 31, 32]. It may of course be possible to avoid divergence in specific applications, or at least slow it down long enough, for example via target networks or replay buffers. Nevertheless, guaranteed convergence has clear advantages. Similar to recent work using policy gradient methods, recent work using dynamic programming methods have adopted multi-layer networks without careful side-by-side comparisons to simpler architectures. Could a global quadratic approximation to the optimal value function (which is linear in the set of quadratic features) be sufficient to solve most of the continuous control tasks currently studied in RL? Given that quadratic value functions correspond to linear policies, and good linear policies exist as shown here, this might make for interesting future work.

# 6 Conclusion

In this work, we demonstrated that very simple policy parameterizations can be used to solve many benchmark continuous control tasks. Furthermore, there is no significant loss in performance due to the use of such simple parameterizations. We also proposed global variants of many widely studied tasks, which requires the learned policies to be competent for a much larger set of states, and found that simple representations are sufficient in these cases as well. These empirical results along with Occam's razor suggests that complex policy architectures should not be a default choice unless side-by-side comparisons with simpler alternatives suggest otherwise. Such comparisons are unfortunately not widely pursued. The results presented in this work directly highlight the need for simplicity and generalization in RL. We hope that this work would encourage future work analyzing various architectures and associated trade-offs like computation time, robustness, and sample complexity.

## Acknowledgements

This work was supported in part by the NSF. The authors would like to thank Vikash Kumar, Igor Mordatch, John Schulman, and Sergey Levine for valuable comments.

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

# A Choice of Step Size

Compare $\alpha$ vs $\delta$ here. An important design choice in the version of NPG presented in this work is normalized vs un-normalized step size. The normalized step size corresponds to solving the optimization problem in equation (3), and leads to the following update rule:

$$\theta_{k+1} = \theta_k + \sqrt{\frac{\delta}{g^T \hat{F}_{\theta_k}^{-1} g}} \; \hat{F}_{\theta_k}^{-1} g.$$

On the other hand, an un-normalized step size corresponds to the update rule:

$$\theta_{k+1} = \theta_k + \alpha \; \hat{F}_{\theta_k}^{-1} g.$$

The principal difference between the update rules correspond to the units of the learning rate parameters $\alpha$ and $\delta$. In accordance with general first order optimization methods, $\alpha$ scales inversely with the reward (note that $F$ does not have the units of reward). This makes the choice of $\alpha$ highly problem specific, and we find that it is hard to tune. Furthermore, we observed that the same values of $\alpha$ cannot be used throughout the learning phase, and requires re-scaling. Though this is common practice in supervised learning, where the learning rate is reduced after some number of epochs, it is hard to employ a similar approach in RL. Often, large steps can destroy a reasonable policy, and recovering from such mistakes is extremely hard in RL since the variance of gradient estimate for a poorly performing policy is higher. Employing the normalized step size was found to be more robust. These results are illustrated in Figure 5

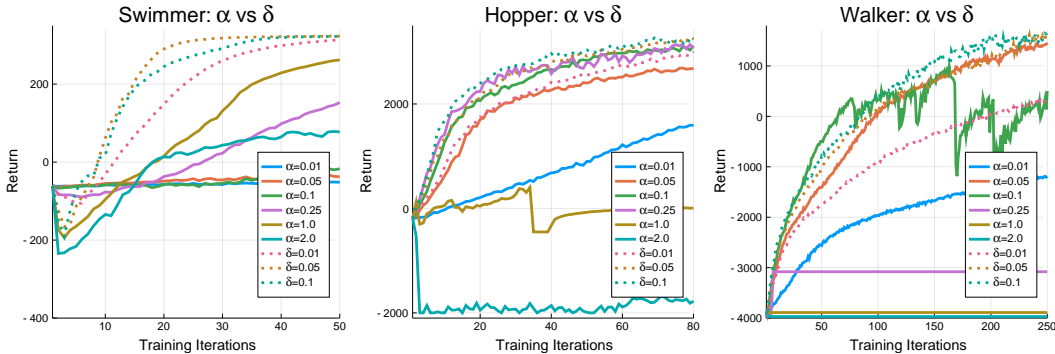

Figure 5: Learning curves using normalized and un-normalized step size rules for the diverse versions of swimmer, hopper, and walker tasks. We observe that the same normalized step size ($\delta$) works across multiple problems. However, the un-normalized step size values that are optimal for one task do not work for other tasks. In fact, they often lead to divergence in the learning process. We replace the learning curves with flat lines in cases where we observed divergence, such as $\alpha = 0.25$ in case of walker. This suggests that normalized step size rule is more robust, with the same learning rate parameter working across multiple tasks.

# B Effect of GAE

For the purpose of advantage estimation, we use the GAE [13] procedure in this work. GAE uses an exponential average of temporal difference errors to reduce the variance of policy gradients at the expense of bias. Since the paper explores the theme of simplicity, a pertinent question is how well GAE performs when compared to more straightforward alternatives like using a pure temporal difference error, and pure Monte Carlo estimates. The $\lambda$ parameter in GAE allows for an interpolation between these two extremes. In our experiments, summarized in Figure 6, we observe that reducing variance even at the cost of a small bias ($\lambda = 0.97$) provides for fast learning in the initial stages. This is consistent with the findings in Schulman et al. [13] and also make intuitive sense. Initially, when the policy is very far from the correct answer, even if the movement direction is not along the gradient (biased), it is beneficial to make consistent progress and not bounce around due to high

variance. Thus, high bias estimates of the policy gradient, corresponding to smaller $\lambda$ values make fast initial progress. However, after this initial phase, it is important to follow an unbiased gradient, and consequently the low-bias variants corresponding to larger $\lambda$ values show better asymptotic performance. Even without the use of GAE (i.e. $\lambda = 1$), we observe good asymptotic performance. But with GAE, it is possible to get faster initial learning due to reasons discussed above.

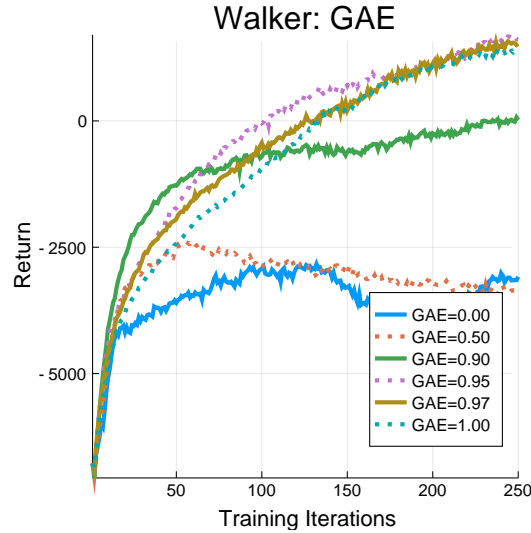

Figure 6: Learning curves corresponding to different choices of $\lambda$ in GAE. $\lambda = 0$ corresponds to a high bias but low variance version of policy gradient corresponding to a TD error estimate: $\hat{A}(s_t, a_t) = r_t + \gamma V(s_{t+1}) - V(s_t)$; while $\lambda = 1$ corresponds to a low bias but high variance Monte Carlo estimate: $\hat{A}(s_t, a_t) = \sum_{t'=t}^{T} \gamma^{t'-t} r_{t'} - V(s_t)$. We observe that low bias is asymptotically very important to achieve best performance, but a low variance gradient can help during the initial stages.

