[Reviews · NeurIPS 2017]

Reviewer 1



This paper presents a policy search with Natural Gradient algorithm using linear and RBF features. It obtains state of the art results (compared to TRPO with neural networks) on OpenAI benchmarks. These results highlight the need for simplicity and generalization in RL. The algorithm can be seen as an actor-critic algorithm where the actor (policy) is updated by a natural gradient step and the critic (value function) is estimated by matching the n-step returns. Then to compute the advantage function they use a GAE procedure. The authors choose a linear and RBF representations of the states to learn the policy. Those representations can also be interpreted as neural networks. So, one way to look at this paper is that the authors come up with simpler neural networks in order to achieve state of the art results on those continuous-control tasks. Thus, it will be very interesting to compare the number of the learnable parameters between this algorithm and TRPO with NN. In addition, it would be nice to add the number of generated trajectories used with this implementation and TRPO to compare the data efficiency of the algorithm. Another point that should be addressed is that there is no clean proof of convergence of Q Learning (at my knowledge) with linear features (except tabular case) towards the optimal action-value function. So I do not think that it is a good argument to use. Finally, I think that the paper raises an interesting point about simplicity of the architectures. However, they should provide the ratio of the number of learnable parameters between their implementation and TRPO and give the number of trajectories used to compare data-efficiency. Those should be good arguments to prefer simpler neural networks architectures.

Reviewer 2



This paper takes an critical view on recent results for deep reinforcement learning. The paper uses a simple natural gradient algorithm (comparable to TRPO) to either learn a linear policy or a policy linear in RBF features generated by a random Fourier basis. Despite the simplicity of the policy architecture, comparable results could be achieved with these linear policies. Moreover, the paper analyses the generalisation ability of the policies by making the task harder (action limits, more difficult and varying initial states and perturbations). The linear architecture using RBF features could learn also these more challenging tasks. This paper does not provide any theoretical or algorithmic contribution. However, it is still an interesting paper as it takes a critical view at the results presented for recent deep reinforcement learning algorithms. It was generally believed that deep architectures are needed to learn such tasks. Hence, the results are quite surprising and interesting. Linear architectures do have several advantages in RL to non-linear techniques. However, linear architectures where thought to be not expressive enough to learn these complex tasks. This does not seem to be true. Even if there are no theoretical or algorithmic contributions in this paper, it gives us the opportunity to rethink the "deep is always good story" for reinforcement learning. If a simple architecture works equally well, it should be preferred. In summary, the paper is interesting and food for thought, but does not contain theoretical or algorithmic contributions. I would rate it nevertheless as borderline, leaning towards accepting the paper.

Reviewer 3



The paper evaluates natural policy gradient algorithm with simple linear policies on a wide range of “challenging” problems from OpenAI MuJoco environment, and shows that these shallow policy networks can learn effective policies in most domains, sometimes faster than NN policies. It further explores learning robust and more global policies by modifying existing domains, e.g. adding many perturbations and diverse initial states. The first part of the paper, while not proposing new approaches, offers interesting insights into the performance of linear policies, given plethora of prior work on applying NN policies as default on these problems. This part can be further strengthened by doing ablation study on the RL optimizer. Specifically, GAE, sigma vs alpha in Eq. 5, and small trajectory batch vs large trajectory batch (SGD vs batch opt). In particular, it is nice to confirm if there are subtle differences from prior NPG implementation with linear policies that made the results in this paper significantly better (e.g. in TRPO [Schulman et. al., 2015], one of the major differences was sigma vs alpha). It could be unsurprising that a global linear policy may perform very well in narrow state space. The second part of the paper tries to address such concern by modifying the environments such that the policies must learn globally robust policies to succeed. The paper then again showed that the RBF/linear policies could learn effective recover policies. It would be great to include corresponding learning curves for these in Section 4 beyond Figure 4, comparing NN, linear and RBF. Are there figures available? Additional comment: - A nice follow-up experiment to show is if linear policies can solve control problems from vision, with some generic pre-trained convolutional filters (e.g. ImageNet). This would further show that existing domains are very simple and do not require deep end-to-end optimization.